# Effects of Different Biochars on Wheat Growth Parameters, Yield and Soil Fertility Status in a Silty Clay Loam Soil

**DOI:** 10.3390/molecules24091798

**Published:** 2019-05-09

**Authors:** Tanveer Ali Sial, Zhilong Lan, Limei Wang, Ying Zhao, Jianguo Zhang, Farhana Kumbhar, Mehurnisa Memon, Muhammad Siddique Lashari, Ahmed Naqi Shah

**Affiliations:** 1College of Natural Resources and Environment, Northwest A&F University, Yangling 712100, China; alisial@nwafu.edu.cn (T.A.S.); zll199101@gmail.com (Z.L.); wanglimei@nwafu.edu.cn (L.W.); 2Department of Soil Science, Sindh Agriculture University, Tandojam 70060, Pakistan; nisamemon@gmail.com (M.M.); mslashari@gmail.com (M.S.L.); 3College of Resources and Environmental Engineering, Ludong University, Yantai 264025, China; 4Department of Biotechnology, Sindh Agriculture University, Tandojam 70060, Pakistan; kumbharfarhana@yahoo.com; 5Department of Agronomy, Sindh Agriculture University, Tandojam 70060, Pakistan; ahmednaqishah@gmail.com

**Keywords:** biochars, root traits, grain yield, soil enzymes activities, soil nutrients status

## Abstract

The conversion of organic wastes into biochar via the pyrolysis technique could be used to produce soil amendments useful as a source of plant nutrients. In this study, we investigated the effects of fruit peels and milk tea waste-derived biochars on wheat growth, yield, root traits, soil enzyme activities and nutrient status. Eight amendment treatments were tested: no amendment (CK), chemical fertilizer (CF), banana peel biochar 1% (BB1 + CF), banana peel biochar 2% (BB2 + CF), orange peel biochar 1% (OB1 + CF), orange peel biochar 2% (OB2 + CF), milk tea waste biochar 1% (TB1 + CF) and milk tea waste biochar 2% (TB2 + CF). The results indicated that chlorophyll values, plant height, grain yield, dry weight of shoot and root were significantly (*p* < 0.05) increased for the TB2 + CF treatment as compared to other treatments. Similarly, higher contents of nutrients in grains, shoots and roots were observed for TB2 + CF: N (61.3, 23.3 and 7.6 g kg^−1^), P (9.2, 10.4 and 8.3 g kg^−1^) and K (9.1, 34.8 and 4.4 g kg^−1^). Compared to CK, the total root length (41.1%), surface area (56.5%), root volume (54.2%) and diameter (78.4%) were the greatest for TB2 + CF, followed by BB2 + CF, OB2 + CF, TB1 + CF, BB1 + CF, OB1 + CF and CF, respectively. However, BB + CF and OB + CF treatments increased β-glucosidase and dehydrogenase, but not urease activity, as compared to the TB + CF amendment, while all enzyme activity decreased with the increased biochar levels. We concluded that the conversion of fruit peels and milk tea waste into biochar products contribute the benefits of environmental and economic issues, and should be tested as soil amendments combined with chemical fertilizers for the improvement of wheat growth and grain yield as well as soil fertility status under field conditions.

## 1. Introduction

Wheat (*Triticum aestivum* L.) is considered as a key nutritional crop for feed throughout the worlds. Winter wheat and summer maize (*Zea mays* L.) are major cereal crops for food production in the Loess Plateau of China [1]. The continuous increase of the global population has had a significant effect on food production and water resources. Agricultural soil fertility is in a constant decline due to a lack of application of organic amendments and the continuous application of chemical fertilizers [2]. In particular, the utilization of nitrogen fertilizers [3], has decreased the level of soil organic carbon (SOC) [4] and low fertilizer use efficiency [5,6] contributes to the decline of soil fertility. In China, farmers apply high doses of chemical fertilizer to obtain high grain yields [7,8], leading to serious environmental issues and economic issues due to high expense of fertilizer inputs [9,10]. The continuous, sole application of chemical fertilizer into agriculture fields causes soil degradation [11] and low crop quality [1,12]. The sole application of chemical fertilizers is not effective for soil organic carbon stock and root traits. These major issues exist globally in farming systems [13], but particularly in the arid farming regions of northwestern China [14]. Meeting the food requirements of future generations will lead to further stresses on soil and water resources [15]. Some previous studies [8,16,17] reported that the co-application of organic amendments with chemical fertilizers increased crop growth, grain yield and root traits in addition to improving soil fertility. Notably, the recycling of nutrients through the conversion of organic wastes into biochar and their application as a soil amendment is a sustainable practice for plant nutrients requirement and root traits [2,4,6]. The root is the main bridge between the aboveground biomass of the plant and soil water, through which the plant uptakes nutrients from the soil medium [4]. This pathway is affected by many environmental issues, viz. water stress, low soil fertility, salt tress and metal toxicity [4,6,15]. Root length (RL) and root surface area (SA) are parameters that dictate the plant’s capacity to uptake soil nutrients [18]. Soil mineral nitrogen (NH_4_^+^-N) is easily taken up by the roots, and NO_3_^−^-N is available in a mobile form in the xylem and can be stored in the vacuoles of shoots and roots [19]. It also plays an important role in the transport of photosynthetically fixed carbon to soil organic matter (SOM) pools and carbon sequestration [20]. In recent decades, the application of biochar to agricultural soils has been proven to be a good practice for climate change and soil carbon sequestration [1,17], and this practice has led to positive effects on root morphology and functioning [21]. Biochar is a carbon rich solid substance produced under a controlled pyrolysis temperature in a negligible O_2_ environment [18,22]. It has higher pore space and surface area [17], higher water holding capacity [23], reduced soil bulk density [10] and reduced nutrients losses [6]. Consequently, the application of biochar could improve soil physical, chemical and biological properties [24], addition to affecting soil carbon and nitrogen cycles [2]. Biochar contains macronutrients [25] and increases soils nutrient availability [26], thus improving plant growth and grain yield [27,28]. Moreover, it improves rainfall retention and reduces water stress in arid and semi-arid regions [29]. Nutrients contents are dependent on the biochar pyrolysis temperature, time and feedstock [2,17,26]. Many studies have reported inconsistent results for crop yield, plant nutrients uptake [1,30,31] and soil properties [10,26]. These variations could be dependent on soil and biochar feedstock properties [32,33]. To date, no results have been reported for biochar generated from fruit peels and milk tea wastes, and applied for wheat growth and root traits in loess soils. The present study assesses soil amendments of biochar derived from fruit peels and milk tea waste for wheat growth, root traits and soil fertility status. The huge quantity of fruit wastes generated every day causes environmental and economic issues due to direct landfilling, but these issues could be mitigated by the use of fruit waste for soil amendment.

It is estimated that some 3.5 million tons of fruit waste (FW) is produced per day in Beijing and Shanghai [34]. Unfortunately, this huge amount of organic waste is usually disposed of by landfilling or burning, which increases greenhouse gas emissions [22,26]. The conversion of organic waste into biochar as a soil amendment is a good way to improve plant growth and nitrogen (N) availability as well as reduce losses by leaching [35], having positive effects on soil phosphorus (P) and potassium (K) availability [26,36]. Recent studies have shown that biochar application increased soil enzymes activities in N and P cycling [17,31] but decreases the activities of carbon (C) related enzymes in soil [22,25]. However, some studies have shown inconsistent results [35,37], depending on the soil type [33] and pyrolysis temperature of the biochar [38]. Soil enzymes are involved in all biochemical reactions [25], metabolism, and the break down and decomposition of organic matter and residues [26]. Furthermore, they accelerate soil nutrients cycling [17,31].

To manage environmental and economic issues as well as to improve soil quality and crop production in loess areas, most studies have focused on crop straw or organic manure-derived biochars assoil amendments in loess plateau areas. There is limited study on the use of biochar derived from the fruit peels and milk tea wastes for wheat growth, root traits and soil fertility status. In our study we used biochar based on fruit peels and milk tea wastes because these contain macro- and micronutrients in addition to serving as a carbon source. These organic materials could be the best alternative source for biochar production. Therefore, we used fruit and milk tea waste derived biochars (at a low pyrolysis temperature of 350 °C) as nutritional sources combined with inorganic fertilizers for the treatment of loess soil. The specific objectives were: (1) to compare the effects of different biochar amendments on wheat growth, yield and root traits; (2) to quantify their effects on the biochemical properties of loess soil.

## 2. Results and Discussion

### 2.1. Scanning Electron Microscopy (SEM)

The morphology of biochars derived from banana and orange peels and milk tea waste was detected from SEM micrographs after the different biochar surfaces were pyrolyzed at a constant pyrolysis temperature of 350 °C for 2 h. The milk tea waste biochar (TB) had a greater surface area and a higher pore space than the banana peel biochar (BB) and orange peel biochar (OB) (Figure 1).

### 2.2. Influence of Soil Amendments on Wheat Growth Parameters

Eight treatments were tested: no amendment (CK), chemical fertilizer (CF), banana peel biochar 1% (BB1 + CF), banana peel biochar 2% (BB2 + CF), orange peel biochar 1% (OB1 + CF), orange peel biochar 2% (OB2 + CF), milk tea waste biochar 1% (TB1 + CF) and milk tea waste biochar 2% (TB2 + CF). The application of different biochar amendments combined with inorganic fertilizers significantly (*p* < 0.05) increased chlorophyll values, plant height (cm) and shoot and root dry weight (g pot^−1^) as compared to CF alone and CK treatments ( Figure 2A–E). The chlorophyll values of wheat leaves were obviously higher in TB2 + CF, BB2 + CF and OB2 + CF (10.12%, 9.35% and 8.64%) than in CK. However, limited difference was observed between CF alone, BB1 + CF, OB1 + CF, OB2 + CF and TB1 + CF treatments, through all of them slightly increased as compared to CK. The performance of TB + CF treatments were generally better compared to those of BB + CF and OB + CF (Figure 2D). As a plant growth indicator, chlorophyll content was significantly (*p* < 0.01) correlated with grain yield, root traits and nutrients contents in grains, shoots and roots (Table 1). Our findings agree well with previous pot experiments [6,13,28]. This might be caused by the higher N content in TB (Table 2).

The biochar amendments apparently (*p* < 0.05) improved the grain yield over the chemical fertilizer alone and control treatments. The grain yield (g pot^−1^) was increased by CF and by co-application with biochar treatments as compared to CK (Figure 2E). The maximum grain yield was recorded for TB2 + CF (9.7 g pot^−1^ or 58.5%) and BB2 + CF (8.4 g pot^−1^ or 51.9%). However, compared to the CK, other treatments also exhibited increased grain yield, BB1 + CF (36.3%), OB1 + CF (27.1%), OB2 + CF (35.3%), TB1 + CF (40.4%) and CF (26.6%). The maximum plant height was recorded for the TB1 + CF and TB2 + CF treatments (48.76 cm and 50.65 cm), followed by the BB2 + CF, OB2 + CF, BB1 + CF, OB1 + CF and CF treatments. The minimum plant height was recorded for CK (36.11 cm). A similar situation was observed for shoot and root dry weight, the highest shoot and root dry weight was recoded for TB2 + CF and BB2 + CF: TB2 + CF /BB2 + CF. Specifically, TB2 + CF/BB2 + CF equaled 5.21 g/4.70 g and 1.10 g/1.03 g for shoots and roots, respectively. Percentage increases in shoot and root dry biomass occurred for BB1 + CF (23.1% and 24.1%), for BB2 + CF (35.3% and 37.0%), for OB1 + CF (15.4% and 21.1%), for OB2 + CF (18.6% and 28.1%), for TB1 + CF (24.9% and 26.9%), for TB2 + CF (41.5 and 41.1%) and for CF (15.5% and 18.4%), compared with CK. Similar results were addressed by previous pot and field experiments using biochar amendments added to different crops [1,6,13,39,40]. In our study, TB + CF showed better performance for plant growth parameters and grain yield as compared to the BB + CF and OB + CF. This could be because TB contains high nutrient contents (N and K) and a lower C:N ratio. Jeffery et al. [32] and Sadaf et al. [2] indicated that the plant growth parameters and grain yield depend on the biochar nutrients composition. The grain yield was higher in TB + CF than in BB + CF and OB + CF. These differences could be explained by the high nutrient contents in TB as compared to BB + CF and OB + CF (Table 2). The C:N ratio of biochars also affects plant growth and grain yield parameters [5,41] a high C:N ratio results in low microbial degradation and a lower nutrients mineralization rate [42,43]. The C:N ratio of TB (14.67) was very low as compared to BB and OB (31.67 and 33.58) (Table 2). TB provides favorable conditions to adsorb soil nutrients for plant growth due to its high pore space and large surface area, as shown by SEM analysis, in comparison to BB + CF and OB + CF.

### 2.3. Effects of Soil Amendments on Nutrients (N, P and K) Contents in Grains, Shoots and Roots of Wheat

Crop nutrients (N, P and K) contents were significantly (*p* < 0.05) improved by the biochar amendments and by CF alone (Figure 3A–I). The higher N content in the grain, shoot and root was increased by different biochar amendments (Figure 3A–C). The N uptake for the grains, shoots and roots was highest for TB2 + CF (61.34, 23.34 and 7.56 g kg^−1^) but lowest for the control (22.69, 13.23 and 3.94 g kg^−1^). The percentage increases in grains, shoots ands root were highest for TB2 + CF (63.0, 43.2 and 47.8%), followed by BB2 + CF (60.3, 37.1 and 40.1%), TB1 + CF (54.4, 35.1 and 39.6%), OB2 + CF (52.7, 29.2 and 35.5%), BB1 + CF (50.7, 27.4 and 34.9%), OB1 + CF (41.8, 18.2 and 32.2%) and CF (39.3, 15.6 and 25.5%) over CK. The P content was markedly (*p* < 0.05) increased for the co-application of BB + CF, OB + CF and TB + CF compared with CK (Figure 3D–F). Similarly, TB + CF increased the P content in grains, shoots and roots as compared to BB + CF, OB + CF, CF and CK. The P content showed the maximum percentage increase in grains, shoots and roots for TB2 + CF (55.0%, 41.4% and 53.8%) and the lowest for CF (32.1%, 13.7% and16.6%) as compared CK. However, biochar rates improved the P content, and higher P content values (as compared to CK) were recorded for TB2 + CF, followed by BB2 + CF, TB1 + CF, OB2 + CF, BB1 + CF, OB1 + CF, and CF in grains, shoots and roots. There was a significant (*p* < 0.05) difference in the K content values between the CF treatment alone and the treatments combined with different biochars (Figure 3G–I). The CF application alone increased the K content for grains by 25.5%, for shoots by 17.2% and for roots by 13.5% as compared to CK. Among the biochar amendments, OB + CF slightly increased the K content in grains, shoots and roots over the CK. Higher K content values were recorded for TB2 + CF showing increases by 54.1% in grains, of 42.9% in shoots and of 49.7% in roots over the CK group (Figure 3G–I). Our results are consistent with previous studies which reported that plant N, P and K contents improved with the co-application of biochar and chemical fertilizer application under pot and field conditions [5,31,44,45,46]. In present study, TB + CF amendments clearly affected N, P and K contents in grains, shoots and roots of wheat as compared to BB + CF, OB + CF and CF alone. This could be due to the maximum concentration of nutrients and the low C:N ratio of TB + CF in comparison to those of BB + CF and OB + CF (Table 2). Many studies reported that the high nutrient contents of biochars improved the plant growth attributes and grain yield [2,32,47], while the yield parameters [41,48], microbial degradation and nutrients mineralization rate depended on the C:N ratio of the amendments [17,42]. Biochar application increased the plant nutrient availability [49,50], owing to its large surface area and high porosity [26,51]. Figure 1 shows that TB has a large surface area and high pore space as compared to BB and OB.

### 2.4. The Impact of Soil Amendments on Root Traits

The data showed that root traits were significantly (*p* < 0.05) influenced by the different biochars and rates (Figure 4A–D). CF improved the total root length and root volume (351 cm and 0.30 cm^3^), but had non-significant effects on surface area and root volume (24.7 cm^2^ and 1.3 cm^3^) in comparison to CK. In contrast, the co-application of different biochars and chemical fertilizers improved the root traits. Among all the treatments, the total root length ranged from 307 to 522 cm (Figure 4A). The maximum root length was found in the TB2 + CF treatment. The surface area was influenced by the biochars with different rates combined with the chemical fertilizer, while, CF alone showed almost no difference compared with the CK (Figure 4B). The surface area improved with the biochar amendments + CF as compared to the CK. The maximum percentage increase was recorded for TB2 + CF and BB2 + CF (56.5% and 49.3%), followed by TB1 + CF (46.7%), OB2 + CF (33.2%), BB1 + CF (33.5%) and OB1 + CF (33.1%), which were all higher than the CK. Similarly, the root volume was lower for CK than for CF for and combined with biochar amendments (Figure 4C). The root volume was improved with increasing biochar rates, and the maximum percentage increase was achieved by the TB + CF amendments (138.7% and 54.2% for TB1 + CF and TB2 + CF), followed by BB + CF (35.5% and 43.4% for BB1 + CF and BB2 + CF) and OB + CF (34.2% and 38.7 for OB1 + CF and OB2 + CF) over CK. The biochar amendments increased the root diameter as compared to CF and CK groups (Figure 4D). TB + CF had the strongest positive effect on the root diameter, being more effective than BB + CF and OB + CF. The higher root diameter was recorded for TB2 + CF (4.77 cm^3^) and the lowest was recorded for the CK (1.03 cm^3^) under all treatment rates. This revealed that biochar or organic amendments with chemical fertilizer improved the crop root traits [31,52,53,54]. Biochar application decreased soil bulk density [10] and improved root growth [55], and induced the maximum nutrient (particularly nitrogen) uptake in plant parts (above and below ground parts) [56,57]. A similar picture was observed in that TB + CF amendments decreased soil bulk density (data not mentioned) and maximum nutrient contents for grains, shoots and roots as compared to the BB + CF and OB + CF amendments (Figure 3). This could be attributed to the improved root traits for TB + CF amendments over the other treatments. There was a significant (*p* < 0.01) correlation between root traits and plant nutrients content (Table 1).

### 2.5. The Influence of Different Biochars on Soil Enzyme Activities

Soil enzymatic activities are ascribed to the properties of amendments and soil because of the perceived behavioral differences in various amendments, and their adsorption and complexation with soil colloids. The β-glucosidase activities were significantly (*p* < 0.05) influenced by different biochar amendments (Figure 5A). The maximum β-glucosidase activity was observed for the OB1 + CF (564 mg p-nitrophenyl kg^−1^ soil h^−1^) and the lowest observed for the CK (268 mg *p*-nitrophenyl kg^−1^ soil h^−1^) treatments. The β-glucosidase activity was decreased with increasing biochar rates for all biochar amendments. The OB + CF treatment increased β-glucosidase activity as compared to BB + CF and TB + CF (Figure 5A), which is in reasonable agreement with recent studies [4,13]. This could be due to the higher C:N ratio for OB + CF and BB + CF as compared to TB + CF. Enzyme activity was accelerated during the hydrolysis of the organic compounds [58], and acted as an energetic source for soil microorganisms, implying different carbon hydrolyzing activities [59]; therefore, it is an indicator of changes in carbon–related soil enzymes [60].

Urease activity was accelerated with increasing biochar rates and it was increased by CF treatment as compared to over CK (Figure 5B). The maximum percentage increase was observed for TB2 + CF (72.2%), followed by BB2 + CF (63.9%), TB1 + CF (61.6%), OB2 + CF (61.2%), BB1 + CF (55.5%), OB1 + CF (49.7%) and CF (28.2%) over CK. Among the different biochar amendments, TB + CF had better performance as compared to BB + CF and OB + CF. The accelerated urease activity with increasing biochar levels was also evidenced under short and long-term experiment studies [10,13,26]. Biochar applications increased a series of N- related enzymes [25,35] and the defense of soil enzymes activities [17]. In the present study, TB + CF exhibited the maximum N content and a low C:N ratio as compared to BB + CF and OB + CF (Table 2). The SEM analysis indicated that TB had a larger surface area and a higher pore space as compared to BB and OB (Figure 1), and soil urease activity showed a significant (*p* < 0.01) positive correlation with soil NH_4_^+^-N and NO_3_^−^-N (Table 1). This could be because of TB increased urease activity more than BB and OB amendments. Jindo et al. [61] and Sial et al. [26] showed that biochar application can increase urease activity.

As compared to CK, soil amendments and rates were found to significantly (*p* < 0.05) impact dehydrogenase activity (Figure 5C). The highest dehydrogenase activity was determined for OB1 + CF (37.0 mg Triphenyl formazan (TPF) kg^−1^ soil h^−1^) and the lowest was found for CK (20.5 mg Triphenyl formazan (TPF) kg^−1^ soil h^−1^). Similarly, the TB + CF amendment decreased dehydrogenase activity as compared to the BB + CF and OB + CF amendments. The OB + CF amendment increased dehydrogenase activity, indicating the slow immobilization of soil organic carbon (SOC) and increased soil microbial activity. Our results are in line with previous findings [13,60] which established that carbon- related soil enzyme activity indicates labile carbon compounds in the biochars for enzymatic reaction. In current study, we observed a negative correlation between soil pH, SOC, NO_3_^−^- N and dehydrogenase activity (Table 1).

### 2.6. Influence of Amendments on Soil PH and SOC

The application of different biochar amendments combined with CF significantly (*p* < 0.05) affected soil pH and soil organic carbon (SOC). Soil pH was slightly increased with increasing biochar rates, but showed non-significant (*p* < 0.05) differences from CK (Figure 6A). The highest pH was measured in the BB2 + CF treatment (8.14) and the lowest pH was observed in the CF treatment (8.0). The soil pH was decreased in all treatments except BB2 + CF and OB2 + CF as compared to CK. A similar trend was observed in previous studies [5,13,25], and soil pH increased with increasing biochar levels [17], but decreased in the chemical fertilizer treatment alone as compared to un-amended soils [25]. Overall, biochar increased soil pH [26,33].

SOC concentrations were influenced by the application of different biochar and rates compared with the CF alone and CK (Figure 6B). The maximum SOC concentrations were measured in TB2 + CF (24.4 g kg^−1^) and minimum were observed in CK (8.8 g kg^−1^). The improved percentage of SOC compared to CK decreased in the order of TB2 + CF (66.3%) > BB2 + CF (59.7%) > TB1 + CF (55.4%) > OB2 + CF (52.7%) > BB1 + CF (48.2%) > OB1 + CF (39.1%) > CF (7.2%). The comparison between the biochars and their rates showed that TB + CF amendments increased SOC concentrations as compared to BB + CF and OB + CF rates. The carbon storage of biochar applications in soils depends on their labile natures [2], in addition to the C:N ratios of biochars [42]. In our study, the TB + CF amendment indicated the higher stability of SOC due to the higher surface area and higher pore space, and the lower C:N ratio as compared to BB + CF and OB + CF. Pokharel et al. [26] and Zhu et al. [62] evaluated that a high surface area and pore space provided favorable conditions to adsorb soluble elements from organic matter (OM) and protect against microorganisms as well as decrease C mineralization [26].

### 2.7. Effect of Soil Amendments on Soil Available N, P and K

Figure 6C,D show that soil NH_4_^+^-N and NO_3_^−^-N concentrations were significantly (*p* < 0.05) affected by the amendments as compared to CK. There was significant influence on NH_4_^+^-N concentrations in the CF group alone with the highest NH_4_^+^-N concentration (6.2 mg kg^−1^) treatment. The co-application of biochar and CF amendments resulted in decreased NH_4_^+^-N concentrations over CK group. The soil NH_4_^+^-N concentration decreased with the increase of biochar rates, with the maximum percentage decrease in BB2 + CF (66.6%) and the minimum in TB1 + CF (18.3%) in comparison to CK. In contrast, the soil NO_3_^−^-N concentrations were greater in biochar treatments as compared to the CF and CK groups. The NO_3_^−^-N concentrations decreased with the increase of the biochar levels, but still remained higher than the concentrations observed in the CF and CK groups. The NO_3_^−^-N concentrations ranged from 11.6 to 20.8 mg kg^−1^ among all treatments; the concentration was highest for TB1 + CF (20.8 mg kg^−1^) and lowest for CK (11.6 mg kg^−1^). Our results are consistent with the incubation study of Wang et al. [25] and pot studies of Sadaf et al. [2] and Khan et al. [13], and as well as field experiments of Haider et al. [30] and Zhang et al. [1]. In the present study, soil NH_4_^+^-N was improved following the treatment of CF alone over, as compared to the biochar and CK treatments. This might be because CF alone decreased the soil pH, which was slightly increased under co-applications of different biochars treatments. Esfandbod et al. [63] and Lan et al. [33] recognized that soil NH_4_^+^-N and NO_3_^−^-N concentrations depend on the pH and types of soil amendments.

The interactive influence of different biochar treatments and CF alone significantly (*p* < 0.05) improved soil available P and K concentrations as compared to CK (Figure 6E,F). The available P and K concentrations of soil that was displaced showed a statistically significant increase with increasing biochar rates. The higher P and K concentrations were recorded in TB2 + CF (75.5 mg kg^−1^ and 240 mg kg^−1^) and the lower concentrations were found in CK (25.1 mg kg^−1^ and 160 mg kg^−1^). Among all biochar treatments, a better performance was observed for the TB + CF treatments in comparison to the BB + CF and OB + CF treatments. Similar outcomes after biochar applications were reported in previous studies conducting short- and long- term experiments [13,28,33]. Sadaf et al. [2] established that soil nutrient availability is related to the type of amendments, surface area and pore space of biochars. TB has a large pore space and surface area, explaining its high adsorption of P and K elements as compared to BB and OB (Figure 1). Biochar application improved the SOC and increased the P and K concentrations [1,26,64]. There was a significant (*p* < 0.01) positive correlation among SOC, available P and K (Table 1).

### 2.8. Relationship between the Plant Nutrients Content NPK, Root Traits, Soil Enzymes and Soil Chemical Properties

The RDA indicated that plant growth parameters, nutrients content, grain yields, root traits and soil chemical properties (SOC, AP and AK) and urease activity had a strong relation with each other, while soil pH, mineral N, glucosidase and dehydrogenase activities had weak relation with plant growth and root parameters (Figure 7). The PCA disposed clear differences in the wheat growth, nutrients content, grains yield and root traits, and soil parameters. All studied parameters such as plant growth, nutrients content, grains yield, root traits and soil biochemical properties were clearly assembled into eight different groups. The first group contained the samples from T1 (CK), the second group from T2 (CF), the third group from T3 (BB1 + CF), the fourth group from T4 (BB2 + CF), the fifth group from T5 (OB1 + CF), the sixth group from T6 (OB2 + CF), the seventh group from T7 (TB1 + CF) and the eighth group from T8 (TB2 + CF). The cumulative variances of contributions reached 77.6% and 31.0% for between wheat growth and soil all parameters (Figure 8).

## 3. Materials and Methods

### 3.1. Collection of Feedstock’s and Biochar Production

Banana and orange peel waste and milk tea waste were respectively collected from the local fruit market of Yangling and a milk tea restaurant at Xi’an, Shaanxi, China, and subsequently shipped to the Soil Physics Laboratory of Northwest A&F University. The banana and orange peels and milk tea waste were washed with tap and distilled water four times. The fruit peels cut into small pieces of around 1 inch and air-dried for 12 days, then dried in an oven at 68 °C for 72 h and ground to pass through a 2 mm sieve. These raw materials were pyrolyzed in a laboratory scale using a Modified Muffle Furnace (FO410C-OP01, Yamato, Kyoto, Japan) system in presence of an N_2_ atmosphere with the temperature raised to 350 °C and maintained at required peak for 2 h to generate the biochars. All the biochars were stored in a desiccator for further use, and their basic properties are listed in Table 2.

### 3.2. Soil Sampling Area

Surface soil (0–20 cm) samples were collected from a long-term experimental station of wheat and maize crop rotation an experimental station at Northwest A&F University (NWAFU), Yangling, Shaanxi, China (34° 20 N, 108° 24 E). This station is classified as an Orthic Anthrosol with a silty clay loam in texture [4]. This region is in a warm temperate zone and is partly cloudy during the monsoon season. The mean annual temperature is around 13 °C, where the average in January is −1.6 °C and that is July is 26 °C. The annual precipitation is 630 ± 10 mm. The collected surface soil samples were mixed to provide a composite sample. The composite sample was kept in polyphenylene bags and shipped to the laboratory, and finally air-dried at room temperature (25 ± 2 °C). The collected composite soil samples were ground and passed through a 2 mm sieve. The basic physicochemical properties of soil used in this experiment are shown in Table 3.

### 3.3. Pot Experiment Setup

The pot experiment was performed with a silty clay loam soil and designed as a randomized complete block, including eight treatments with three replicates under an artificial shed. The pot experiment included a control with no amendment (CK), chemical fertilizers (CF), and two levels of soil amendments with biochar derived from banana and orange peels and milk tea waste. In total, the eight treatments included (1) CK (control, no amendment), (2) CF (chemical fertilizers), (3) BB1 + CF (1%; 50 g for 5 kg soil + CF), (4) BB2 + CF (2%; 100 g for 5 kg soil + CF), (5) OB1 + CF (1%; 50 g for 5 kg soil + CF), (6) OB2 + CF (2%; 100 g for 5 kg soil + CF), (7) TB1 + CF (1%; 50 g for 5 kg soil + CF) and (8) TB2 + CF (2%; 100 g for 5 kg soil + CF). The chemical fertilizers nitrogen (N), phosphorus (P) and potassium (K) were applied as a source of urea, di-ammonium and sulfate of potash at the doses of 120, 80 and 60 kg ha^−1^. For each treatment, 5 kg of soil and the proper amendments were mixed manually and added into each pot according to the treatments. After amendment incorporation, the soil in each pot was moistened to about 70% water holding capacity (WHC) by adding tap water and 10 wheat seeds were sown in each pot. The all pots were properly arranged in a completely randomized block design under an artificial shed. The water loss checked every day and maintained a 70% WHC on daily basis up to wheat harvested. After 12 days, wheat plants were thinned and five plants were left in each pot until maturity to assess the grain yield. The wheat plants were uprooted from each pot, and the parameters of aboveground biomass and roots were recorded.

### 3.4. Plant Growth and Yield Parameters

Wheat growth, root traits and grain yield parameters such as chlorophyll values, plant height, biological and grain yields (g pot^−1^) and root biomass were recorded. After the wheat crop was harvested at maturity, the biological yield was recorded by weighing the aboveground biomass of each replication of treatments. The shoots and roots were washed with running water and then washed with distilled water for the removal of adhered soil and biochar particles. Both shoots and roots were accurately dried with tissue paper to record the plant height as well as the shoot and root weight. They were then dried in an oven for 72 h at 68 °C until a constant weight was obtained. Cleaned wheat roots were used for root traits examination including total root length, root surface area, root volume and root diameter, using a Hewlett Packard scanner controlled by Win-RHIZO, 2007d (Reagent Instruments Inc. Ltd., Model–J221 A, Seiko Epson Corporation, Kyoto, Japan) software, according to [26].

### 3.5. Physicochemical Analysis of Soil, Biochar and Plant

Soil electrical conductivity (EC) and pH were measured in an aqueous soil extract in distilled water (1:2.5 (*w*/*v*) soil-water extract), and biochar was applied in a 1:10 (biochar:water) ratio after 30 minutes of shaking at 180 rpm. Soil and biochar EC and pH were measured using a glass electrode (DDS-11AW EC meter) and a pH meter (Mettler-Toledo 320-S, Shanghai Bante Instrument Co., Shanghai, China). Soil particle size was measured by using a Mastersizer 2000E (Malvern, UK) laser diffractometer. Total organic carbon (TOC), total nitrogen (TN) and C:N ratio of soil and biochars were determined using a CN analyzer (Vario Max, Elementar, Norderstedt, Germany). Soil organic carbon (SOC) was measured following wet digestion with sulfuric acid and potassium dichromate (H_2_SO_4_ + K_2_Cr_2_O_7_). Total nitrogen (TN), phosphorus (TP) and potassium (TK) contents of biochar and wheat grains, shoots and roots were analyzed colorimetrically after digestion with H_2_SO_4_ + HClO_4_ [65]. Soil mineral nitrogen (NH_4_^+^-N and NO_3_^−^-N) for soil subsamples was extracted with 2 M of potassium chloride solution (KCl) with a 1:10 (soil: solution) ratio which was shaken for 1 h to determine the concentration of soil NH_4_^+^-N and NO_3_^−^-N using a continuous flow analyzer (Bran and Luebbe AA3, Norderstedt, Germany). Soil available phosphorus (AP) was determined using a 0.5 M NaHCO_3_ with a pH 8.5 extract followed by a visible light spectroscopic analysis of a blue colored complex using a UV-VIS spectrophotometer (Model UV-2450, Shimadzu, Kyoto, Japan) [66]. Soil available potassium (AK) was determined using a 1 N ammonium acetate (NH_4_OAc) extraction followed by emission using an atomic absorption spectrometer (Model Pin AAcle 900F, Perkin Elmer, Santa Clara, CA, USA). Total concentrations of zinc (Zn), iron (Fe) and copper (Cu) in biochar samples were determined by acid digestion using nitric perchloric acid (HNO_3_-HClO_4_; 3:1), and DTPA solution used for soil samples was determined for Zn, Fe and Cu concentrations using an atomic absorption spectrometer. Soil enzyme activities of β-glucosidase, urease and dehydrogenase were analyzed by using the colorimetric method and using (UV-VIS spectrophotometer, Model UV-2450) [67], as briefly explanation described in our previous studies [17,26]. The morphology of biochars was observed by scanning electron microscopy (SEM) micrographs of different biochars surfaces pyrolyzed at 350 °C according to our previous study [4], and the morphological properties are shown in Figure 1.

### 3.6. Statistical Analysis

All experimental data were analyzed using Microsoft Excel 2010 and SPSS 22 software (SPSS Inc., Chicago, IL, USA), and all figures were produced using Origin Pro. 9.0 software (Northampton, MA, USA). The pot experiment results were presented as the mean of triplicates. One-way analysis of variance (ANOVA) was used to evaluate the effects of the treatments. Mean comparison testing was performed using least significant difference (LSD) (*p* > 0.05). Patterns in the plant and soil data were investigated using a bivariate Pearson correlation test. CANOCO 5.0 software was used for redundancy analysis (RDA) and principle component analysis (PCA) to assess the overall differences between the chemical fertilizers alone and combined with different biochars, and to investigate the relationship between plant growth and grain yield parameters/root traits, as well as and the relationship among soil biochemical properties.

## 4. Conclusions

Our study showed that the conversion of banana and orange peels and milk tea waste into biochar products may offer an efficient technique for reducing the environmental and economic issues associated with the disposal of these waste materials. Biochar is the best organic amendment to act as a soil nutrient source to improve wheat growth, grain yield and environmental advantages. The co-application of different biochars and chemical fertilizers significantly improved the wheat growth, root traits and soil fertility as compared to the CF and CK groups. However, TB application had a more significant effect on plant and soil parameters in comparison to BB and OB. Although CF alone increased wheat grain yield and soil N, P, and K contents (as compared to CK), but no significant effect on SOC was observed after wheat harvest. We recommend that the co-application of biochars with chemical fertilizers should be tested in long-term field experiments for the development of soil management strategies, maintained the soil fertility status for the reduction of economic and environment issues.

## Figures and Tables

**Figure 1 molecules-24-01798-f001:**
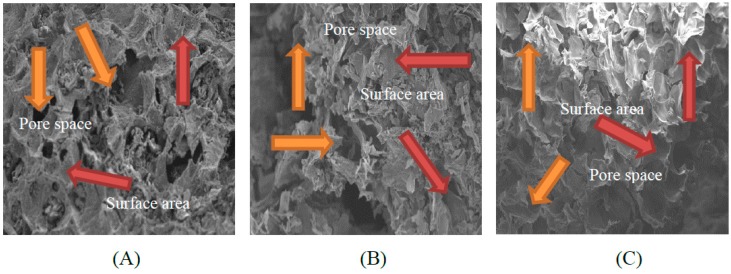
Scanning electron micrographs of banana peels derived biochar (**A**), orange peels derived biochar (**B**) and milk tea waste derived biochar (**C**). The solid orange arrows show the pore space and the red arrows show the surface area.

**Figure 2 molecules-24-01798-f002:**
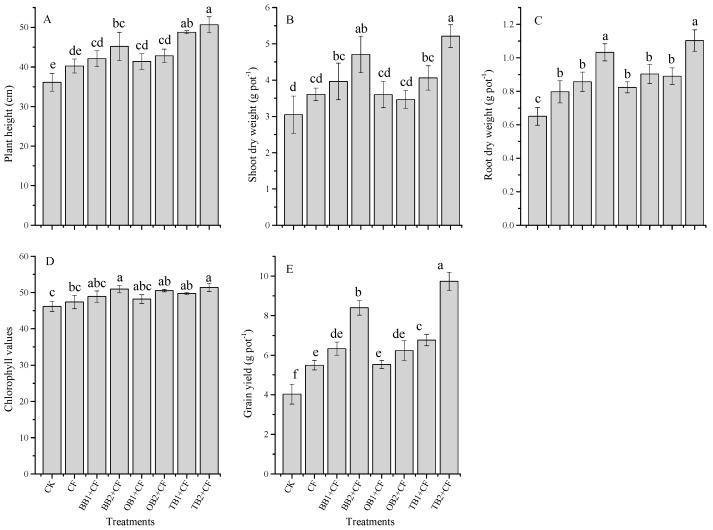
Effect of co-application of biochars and chemical fertilizers on plant height (**A**), shoot dry weight (**B**), root dry weight (**C**), chlorophyll values (**D**) and grain yield (**E**) of wheat. Without amendment (CK), chemical fertilizers (CF), banana peels derived biochar 1% + CF (BB1 + CF), banana peels derived biochar 2% + CF (BB2 + CF), orange peels derived biochar 1% + CF (OB1 + CF), orange peels derived biochar 2% + CF (OB2 + CF), milk tea waste derived biochar 1% + CF (TB1 + CF) and milk tea waste derived biochar 2% + CF (TB2 + CF). Error bars represent the standard deviation of the mean (n = 3). Different letters show there were significant difference (*p* < 0.05) in the LSD means comparisons between the biochars and rates.

**Figure 3 molecules-24-01798-f003:**
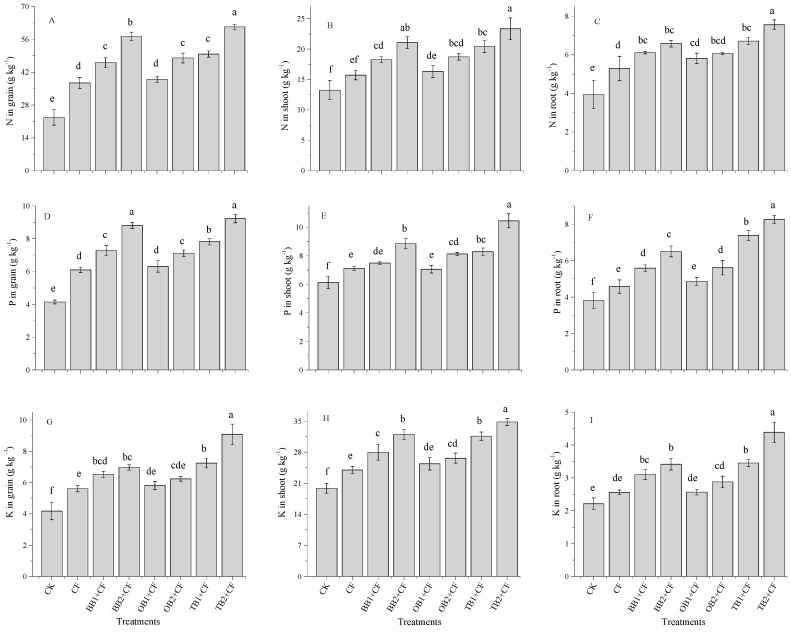
Effect of co-application of biochars and chemical fertilizers on plant nutrients uptake such as N in grain, shoot and root (**A**–**C**), P in grain, shoot and root (**D**–**F**) and K in grain, shoot and root (**G**–**I**) of wheat crop. Without amendment (CK), chemical fertilizers (CF), banana peels derived biochar 1% + CF (BB1 + CF), banana peels derived biochar 2% + CF (BB2 + CF), orange peels derived biochar 1% + CF (OB1 + CF), orange peels derived biochar 2% + CF (OB2 + CF), milk tea waste derived biochar 1% + CF (TB1 + CF) and milk tea waste derived biochar 2% + CF (TB2 + CF). Error bars represent the standard deviation of the mean (n = 3). Different letters show there were significant difference (*p* < 0.05) in the LSD means comparisons between the biochars and rates.

**Figure 4 molecules-24-01798-f004:**
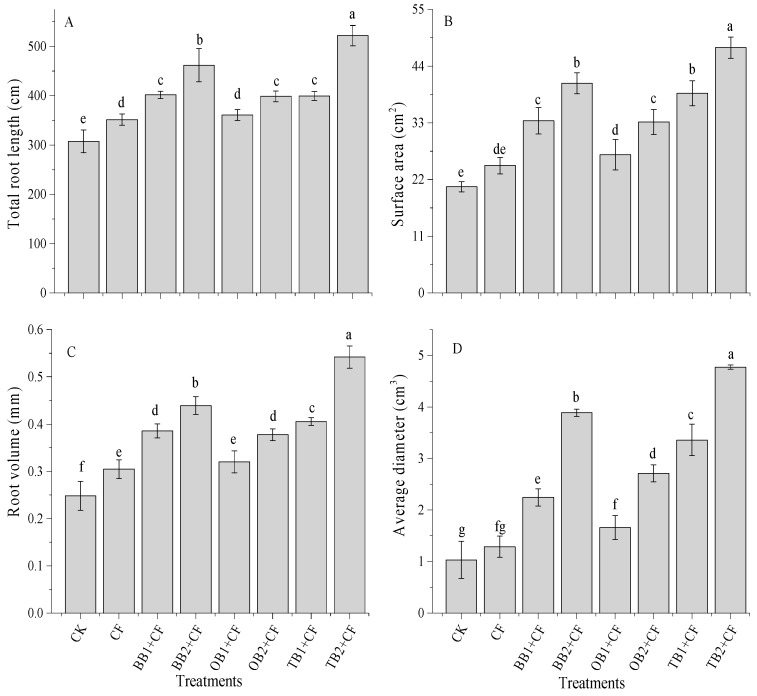
Effect of co-application of biochars and chemical fertilizers on total root length (**A**), surface area (**B**), root volume (**C**) and average diameter (**D**) of wheat. Without amendment (CK), chemical fertilizers (CF), banana peels derived biochar 1% + CF (BB1 +CF), banana peels derived biochar 2% + CF (BB2 + CF), orange peels derived biochar 1% + CF (OB1 + CF), orange peels derived biochar 2% + CF (OB2 + CF), milk tea waste derived biochar 1% + CF (TB1 + CF) and milk tea waste derived biochar 2% + CF (TB2 + CF). Error bars represent the standard deviation of the mean (n = 3). Different letters show that there were significant difference (*p* < 0.05) in the LSD means comparisons between the biochars and rates.

**Figure 5 molecules-24-01798-f005:**
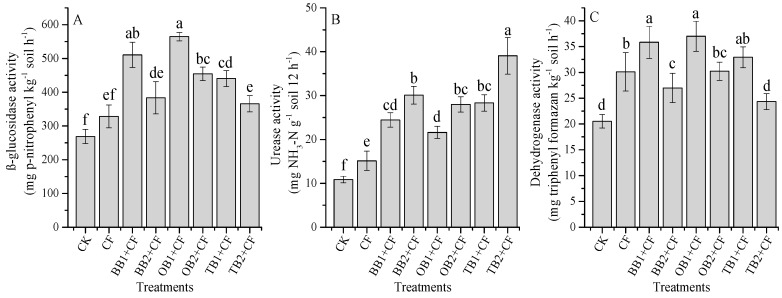
Effect of co-application of biochars and chemical fertilizers on soil enzymatic activities after wheat harvested: β-glucosidase (**A**), urease (**B**) and dehydrogenase (**C**). Without amendment (CK), chemical fertilizers (CF), banana peels derived biochar 1% + CF (BB1 + CF), banana peels derived biochar 2% + CF (BB2 + CF), orange peels derived biochar 1% + CF (OB1 + CF), orange peels derived biochar 2% + CF (OB2 + CF), milk tea waste derived biochar 1% + CF (TB1 + CF) and milk tea waste derived biochar 2%+ CF (TB2 + CF). Error bars represent the standard deviation of the mean (n = 3). Different letters show that there were significant difference (*p* < 0.05) in the LSD means comparisons between the biochars and rates.

**Figure 6 molecules-24-01798-f006:**
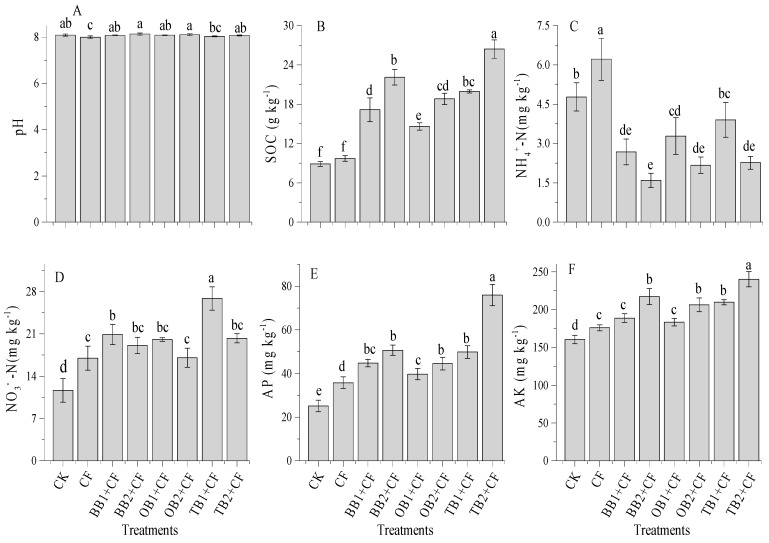
Effect of co-application of biochars and chemical fertilizers on soil pH (**A**), soil organic carbon (**B**), ammonium nitrogen (**C**), nitrate nitrogen (**D**), available phosphorus (**E**) and available potassium (**F**) after wheat harvested. Without amendment (CK), chemical fertilizers (CF), banana peels derived biochar 1% + CF (BB1 + CF), banana peels derived biochar 2% + CF (BB2 + CF), orange peels derived biochar 1% + CF (OB1 + CF), orange peels derived biochar 2% + CF (OB2 + CF), milk tea waste derived biochar 1% + CF (TB1 + CF) and milk tea waste derived biochar 2% + CF (TB2 + CF). Error bars represent the standard deviation of the mean (n = 3). Different letters show there were significant difference (*p* < 0.05) in the LSD means comparisons between the biochars and rates.

**Figure 7 molecules-24-01798-f007:**
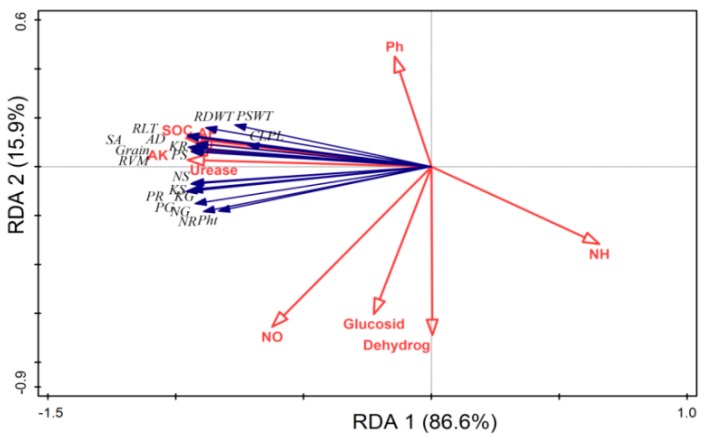
Ordination plot of redundancy analysis (RDA) showing relationships between plant and soil parameters after wheat harvested. Abbreviations = Plant height (Pht), plant shoot dry weight (PSWT), root dry weight (RDWT), chlorophyll SPAD (CLP), nitrogen content in grain (NG), nitrogen content in shoot (NS), nitrogen content in root (NR), phosphorus content in grain (PG), phosphorus content in shoot (PS), phosphorus content in root (PR), potassium content in grain (KG), potassium content in shoot (KS), potassium content in root (KR). Root length (RLT), root surface area (SA), root volume diameter (RVM) and root average diameter (RAVD). Soil organic carbon (SOC), ammonium nitrogen (NH_4_^+^-N), nitrate nitrogen (NO_3_^−^-N), available phosphorus (AP), available potassium (AK), β-glucosidase and dehydrogenase.

**Figure 8 molecules-24-01798-f008:**
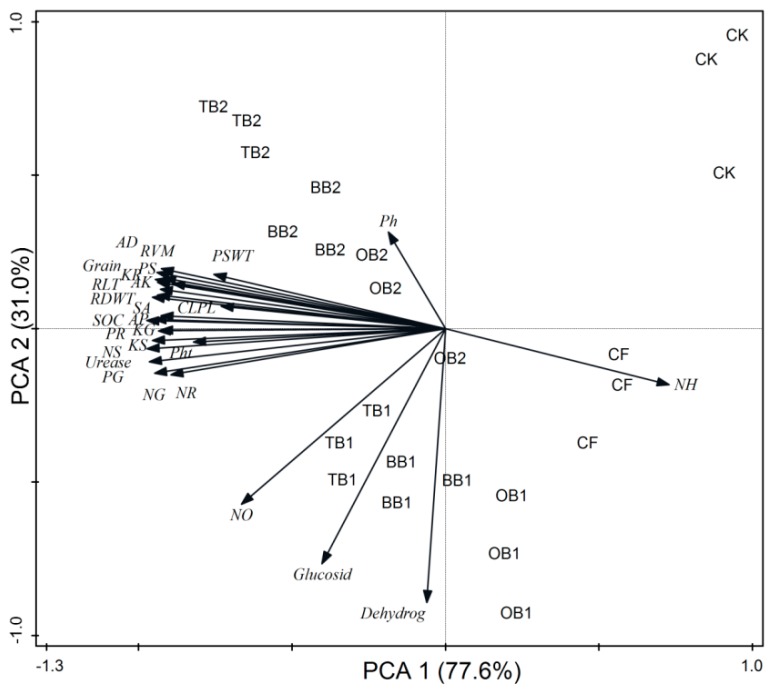
Principle component analysis (PCA) of plant growth parameters, root traits and soil biochemical properties under different soil amendments after wheat crop harvested. These abbreviations are the same with Figure 7.

**Table 1 molecules-24-01798-t001:** The bivariate correlation test between soil chemical and soil enzyme activities after wheat harvested.

	pH	SOC	NH_4_^+^-N	NO_3_^−^-N	AP	AK	β-glucosid.	Urease	Dehydroge.
pH	1	0.255	−0.546 **	−0.134	0.062	0.195	0.037	0.200	−0.284
SOC		1	−0.749 **	0.539 **	0.909 **	0.923 **	0.247	0.946 **	−0.150
NH_4_^+^-N			1	−0.174	−0.559 **	−0.640 **	−0.416 *	−0.691 **	0.094
NO_3_^−^-N				1	0.473 *	0.492 *	0.469 *	0.531 **	0.385
AP					1	0.905 **	0.127	0.886 **	−0.147
AK						1	0.100	0.898 **	−0.162
β-glucosid.							1	0.272	0.633 **
Urease								1	−0.128
Dehydroge.									1

Abbreviations = Soil organic carbon (SOC), ammonium nitrogen (NH_4_^+^-N), nitrate nitrogen (NO_3_^−^-N), available phosphorus (AP), available potassium (AK), β-glucosidase and dehydrogenase. * Correlation is significant at the 0.01 level (2-tailed). ** Correlation is significant at the 0.01 level (2-tailed).

**Table 2 molecules-24-01798-t002:** Basic chemical properties of biochars used in the pot study (mean ± S.E, n = 3).

Parameters	Banana Peels Waste Derived Biochar	Orange Peels Waste Derived Biochar	Milk Tea Waste Derived Biochar
pH (1:10 *v*/*w*) H_2_O	9.85 ± 0.5	8.95 ± 0.4	8.72 ± 0.5
EC (1:10 *v*/*w*) H_2_O	1985 ± 48.5	878 ± 18.5	2055 ± 35.5
Total organic carbon (%)	62.23 ± 6.5	73.58 ± 5.5	71.90 ± 6.2
Total nitrogen (%)	1.97 ± 0.2	2.22 ± 0.5	5.23 ± 0.6
C:N	31.58 ± 4.5	33.14 ± 2.5	13.74 ± 1.2
Total phosphorus (%)	1.15 ± 0.02	0.58 ± 0.02	0.62 ± 0.02
Total potassium (%)	0.39 ± 0.0	0.04 ± 0.0	0.68 ± 0.03
Total zinc (mg kg^−1^)	40.84 ± 3.5	24.92 ± 3.6	39.33 ± 3.5
Total copper (mg kg^−1^)	10.30 ± 1.5	6.29 ± 0.5	11.82 ± 1.2
Total iron (mg kg^−1^)	150.98 ± 6.5	162.92 ± 9.3	533.99 ± 12.5
Yield (%)	42.5 ± 3.5	34.3 ± 2.5	40.5 ± 3.5
Ash content (%)	8.4 ± 1.2	7.2 ± 1.2	9.1 ± 1.2

**Table 3 molecules-24-01798-t003:** Initial physicochemical properties of soil used in the pot study (mean ± S.E, n = 3).

Items	Values
Soil texture	Silty clay loam
pH (1:2.5, H_2_O)	8.15 ± 0.6
EC(1:2.5, H_2_O)	195 ± 6.5
Total carbon (%)	1.97 ± 0.01
Total nitrogen (%)	0.18 ± 0.0
C:N	10.83 ± 0.7
Soil organic carbon (g kg^−1^)	10.05 ± 0.8
Available phosphorus (mg kg^−1^)	23.5 ± 2.5
Exchangeable potassium (mg kg^−1^)	190 ± 5.5
DTPA zinc (mg kg^−1^)	3.7 ± 0.4
DTPA copper (mg kg^−1^)	1.4 ± 0.05
DTPA iron (mg kg^−1^)	7.4 ± 0.8

DTPA = Diethylenetriaminepentaacetic acid.

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
