# Peer review of "Effects of Different Biochars on Wheat Growth Parameters, Yield and Soil Fertility Status in a Silty Clay Loam Soil"

_molecules, 2019, doi:10.3390/molecules24091798_

Round 1
Reviewer 1 Report
General Comment: The results presented in this manuscript are interesting and wroth to be published. However, there are some major issues that need to be answered before this manuscript can be accepted. First, the novelty of this work is not clear. The authors should state clearly how this work can improve the current scientific knowledge, especially the uniqueness of the biochar feedstock selected. In addition, the results are mostly presented superficially. It would be expected that some more in-deep discussion and explanation on the results are given.
Detail Comment:
line 83: The term "milk tea wastes" is confusion. It is referred to as the tea leaves?
Line 110: The term "surface area" is not clear here.
Line 438: What is a "complete block"?
Line 438: "tree replicates"?
line 511: there will be only one "best" fertilizer recommended.
Author Response
REVIEWER: 1Comments and Suggestions for Authors
General Comment: The results presented in this manuscript are interesting and wroth to be published. However, there are some major issues that need to be answered before this manuscript can be accepted. First, the novelty of this work is not clear. The authors should state clearly how this work can improve the current scientific knowledge, especially the uniqueness of the biochar feedstock selected. In addition, the results are mostly presented superficially. It would be expected that some more in-deep discussion and explanation on the results are given.
Comments 1:
First, the novelty of this work is not clear. The authors should state clearly how this work can improve the current scientific knowledge, especially the uniqueness of the biochar feedstock selected.
Response: Thanks a lot for valuable suggestions; we have described novelty of this work (Page No. 2 and line No. 90-95).
Comments 2: Introduction must be improved.
Response: Thank you for the suggestion; we have thoroughly revised the introduction with new additions at several places in order to address the aforementioned points (Page No. 1, line No. 69-76).
Comments 3: Research design can be improved.
Response: We have improved research design.
Comments 4: The methods can be improved described.
Response: Thank you for the suggestion; we have revised the materials and methodology section.
Comments 5: The results can be improved.
Response: The result and discussion have been improved with new literature and well organized as comments.
Comments 6: The conclusions can be improved.
Response: Thanks for your pertinent suggestion; we have re written the conclusion (Page No. 18-19, line No. 578-591).
Reviewer 2 Report
There are many minor corrections to be made.
Line 132 should have TB1 + CF, not TB1
Line 134 should have OB2 + CF, not OB2
Line 135 Change to "not significant"
Lines 136-138 The sentence starting with "The highest ..." needs to be re-written because it is incomprehensible
Line 140 should be For
Line 142 has an "s"
Why do the tables start with Table 4? Also, there is no Table 3
Line 192 should not have a (
Line 196 What is yield parameters?
In Fig. 7 & 8 some of the abbreviations within the figure seem to be different to those under the figure or seem to be missing under the figure
Lines 269/270 have an incorrect explanation. The C:N ratio of OB and BB are not significantly different
Lines 270-273 starting with "The b-glucosidase ..." is difficult to understand and the sentence needs to be re-written
Line 274 should not have a )
Lines 279-281 mentions "biochar" at both the beginning and end of the sentence, which is unclear and needs to be re-written
Line 282 states that TB increased urease activity compared to BB and OB due to a larger surface area, but this is speculation only
Line 288 What does TPF stand for?
Lines 290/291 Why is there information in brackets after OB1 + CF and OB2 + CF?
Lines 291/292 have an incorrect explanation. The C:N ratio of OB and BB are not significantly different
Line 301 Fig. 5C There should not be brackets around "dehydrogenase activity"
Line 336 Table 4 In rows AP and AK there are values with only one *
Line 352 should have CF, not +CF
Lines 368/369 have an incorrect claim. BB contains more P than TB and OB has a similar amount
Line 370 The reference has no number
Line 414 requires superscript o
Line 423 coordinates require superscripts
Line 432 Table 1 under "Items" states pH twice (should it have EC?)
Line 432 Table 1 What is DTPA? Line 488 states DTAPA, so are they the same? Is one spelled incorrectly?
Line 438 should be "three" not "tree"
Line 465 has the reference number in red
Line 476 should have Cr2
Line 487 Perchloric should be one word, and the formula should have a small "l"
Author Response
Comments and Suggestions for Authors
Major comment
Comments 1: The results can be improved.
Response: The result and discussion have been improved with new literature and well organized as comments.
There are many minor corrections to be made.
Comments 1: Line 132 should have TB1 + CF, not TB1
Response: Corrected
Comments 2: Line 134 should have OB2 + CF, not OB2
Response: Corrected
Comments 3: Line 135 Change to "not significant"
Response: We have changed as per suggestion
Comments 4: Lines 136-138, the sentence starting with "The highest ..." needs to be re-written because it is incomprehensible.
Response: Thanks for suggestion, we have rewritten full sentence.
Comments5: Line 140 should be For
Response: As per suggestion have changed.
Comments 6: Line 142 has an "s"
Response: Deleted s
Comments 7: Why do the tables start with Table 4? Also, there is no Table 3.
Response: We have corrected the table’s orders according to position and deleted the text Table 3 which was mistake.
Comments 8: Line 192 should not have a (
Response: Deleted as per suggestion
Comments 9: Line 196 What is yield parameters?
Response: Thanks a lot for comments; we have mentioned the yield parameters throughout paper.
Comments 10: In Fig. 7 & 8 some of the abbreviations within the figure seem to be different to those under the figure or seem to be missing under the figure.
Response: As per comments we have been mentioned all abbreviations which included in the figure 7 and 8.
Comments 11: Lines 269/270 have an incorrect explanation. The C:N ratio of OB and BB are not significantly different.
Response: Thank you very much for fruitful suggestions, we have corrected the explanation.
Comments 12: Lines 270-273 starting with "The b-glucosidase ..." is difficult to understand and the sentence needs to be re-written.
Response: We have re-written full sentence.
Comments 13: Line 274 should not have a )
Response: Deleted as per suggestion.
Comments 14: Lines 279-281 mentions "biochar" at both the beginning and end of the sentence, which is unclear and needs to be re-written.
Comments 15: Line 282 states that TB increased urease activity compared to BB and OB due to a larger surface area, but this is speculation only.
Response: We have re-written the statement.
Comments 16: Line 288, what does TPF stand for?
Response: We have explained the full form of TPF (Triphenyl formazan) in figure.
Comments 17: Lines 290/291 Why is there information in brackets after OB1 + CF and OB2 + CF?
Response: As per suggestion, we have deleted the brackets.
Comments 18: Lines 291/292 have an incorrect explanation. The C:N ratio of OB and BB are not significantly different.
Response: Thanks for valuable comments, we have corrected the explanation.
Comments 19: Line 301 Fig. 5C There should not be brackets around "dehydrogenase activity"
Response: Deleted the brackets from Fig. 5C.
Comments 20: Line 336 Table 4 In rows AP and AK there are values with only one *
Response: Corrected
Comments 21: Line 352 should have CF, not +CF
Response: Corrected this kind of mistake in whole manuscript.
Comments 22: Lines 368/369 have an incorrect claim. BB contains more P than TB and OB has a similar amount.
Response: Thanks for your pertinent suggestion, we have corrected and re-written.
Comments 23: Line 370, The reference has no number.
Response: Mentioned the reference number.
Comments 24: Line 414 requires superscript o.
Response: Corrected.
Comments 25: Line 423 coordinates require superscripts.
Response: Corrected.
Line 432 Table 1 under "Items" states pH twice (should it have EC?).
Comments 26: Line 432 Table 1 What is DTPA? Line 488 states DTAPA, so are they the same? Is one spelled incorrectly?
Response: Thanks, we have written the EC.
Comments 27: Line 438 should be "three" not "tree".
Response: Corrected.
Comments 28: Line 465 has the reference number in red.
Response: Corrected.
Comments 29: Line 476 should have Cr2.
Response: Corrected.
Comments 29: Line 487 Perchloric should be one word, and the formula should have a small "l"
Response: As per suggestion, we have corrected.
Reviewer 3 Report
I will be blunt, before I labor into this process, the paper must be withdrawn and submitted for revision by a Chinese to English professional translator. It is my understanding MDPI offers this service. I have reviewed some excellent results from it! The revision process will benefit you greatly.
I had questions with the current paper after I found the following while quickly scanning the abstract: "We concluded that all biochar products could be used to combine with chemical fertilizers for improvement of wheat growth and grain yield as well as soil fertility status under field conditions."
I confirmed this suspected translation issue when I read the accurate summation within Lines 58 - 69 that the results of biochar studies depend on a number of factors including feedstock, pyrolysis temperature, time, and soil properties. I am sure that buried somewhere in your study there is a basis to conclude that this particular biochar has potential benefit when managed within an appropriate and complete nutrient management scheme. I am not so sure about "all biochar products".
I did not need to read very much to find another example showing the need for more translation.
From the Introduction:
The second sentence (from Lines 46 - 48 is not only inaccurate in English:
"However, the agricultural production is constantly declined due to high dose of chemical fertilizers [2], especially nitrogen fertilizers [3], and low organic amendments decrease soil organic carbon (SOC) [4] and fertilizer use efficiency [5,6]."
But apparently contradicted by the next sentence (Lines 49 - 50):
"In China, farmers always applied high dose of chemical fertilizer to obtain high grain yield [7,8], leading to serious environment and economic issues [9,10]."
Is the second sentence referring to the global situation and the third referring to China specifically? Or is the second referring to the diminishing yield return to very high fertilizer rates, and the third saying that Chinese farmers ignore those diminishing returns despite the negative economics and environmental side-effects?
So within the span of Lines 35 - 57, there are two situations that would benefit from more preciseness - therefore my rapid review. I will be glad to look at the revised draft.
Author Response
REVIEWER: 3Comment 1. English language and style (x) Extensive editing of English language and style required.
Response: Along with the revision, we have improved the English language and style from MDPI English editing company.
Comment 2.The introduction must be improved.
Response: Thank you for the suggestion; we have thoroughly revised the introduction with new additions at several places in order to address the aforementioned points.
Comments 3: Research design can be improved.
Response: We have improved research design.
Comments 4: The methods must be improved.
Response: Thank you for the suggestion; we have revised the materials and methodology section.
Comments 5: The results must be improved.
Response: The result and discussion have been improved with new literature and well organized as comments.
Comments 6: The conclusions must be improved.
Response: Thanks for your pertinent suggestion; we have re written the conclusion (Page No. 18-19, line No. 578-591).
Comments 7: I had questions with the current paper after I found the following while quickly scanning the abstract: "We concluded that all biochar products could be used to combine with chemical fertilizers for improvement of wheat growth and grain yield as well as soil fertility status under field conditions."
Response: As fruitful advised, we have thoroughly revised abstract and improved it.
Comments 8: The second sentence (from Lines 46 - 48 is not only inaccurate in English:
"However, the agricultural production is constantly declined due to high dose of chemical fertilizers [2], especially nitrogen fertilizers [3], and low organic amendments decrease soil organic carbon (SOC) [4] and fertilizer use efficiency [5,6]."
Response: Thanks a lot for your technical suggestions, we have re-written sentence and improved English language throughout the manuscript.
Comments 8: "In China, farmers always applied high dose of chemical fertilizer to obtain high grain yield [7,8], leading to serious environment and economic issues [9,10]."
Is the second sentence referring to the global situation and the third referring to China specifically? Or is the second referring to the diminishing yield return to very high fertilizer rates, and the third saying that Chinese farmers ignore those diminishing returns despite the negative economic and environmental side-effects?
Response: Thanks a lot for your valuable comments, we have re-written the introduction section and clearly mentioned current issues.
Round 2
Reviewer 3 Report
Apparently you uploaded an inter-author review draft with "Track Changes" enabled instead of a final paper. I assume this was a mistake as I am not a co-author. My role is review of your final product. Because this is not a final product, I will not review it further than the following:
The best I can understand from what I was sent is that the English is still problematic for an international audience.
The last sentence of the abstract appears to be a random comment rather than based on this work.
Author Response
Ms. Genie Lu
Section Managing Editor
Molecules
Subject: Minor Revision submission of Manuscript Molecules-481116
We are grateful to the editorial team and the reviewers for their critical evaluation and giving us the opportunity to revise our manuscript. We found all the questions and suggestions raised by the reviewers valuable for our manuscript. We have thoroughly revised our manuscript following the suggestions of reviewers. The point-by-point responses to all the comments and suggestions of reviewer 3 are well prepared and available at the end of the letter and also uploaded the clean copy of manuscript according to the suggestion of reviewer.
New additions in the revised version of manuscript are highlighted with track changes. However the changes made in response to reviewer 3 highlighted with (Color). English editing changes have changed throughout paper respectively. We hope that the present version of manuscript and our accompanying responses will be sufficient to make it suitable for publication in your august journal Molecules.
We shall look forward to hearing from you at your earliest convenience.
Thank you very much for consideration of our manuscript.
With Profound Regards,
Professor Ying Zhao & Professor Zhang Jianguo
yzhaosoils@gmail.com; zhangjianguo21@nwafu.edu.cn
College of Natural Resources and Environment,
Northwest A & F University, Yangling,
Shaanxi 712100, China.
REVIEWER: 3
Comments and Suggestions for Authors
Comments 1:
Apparently you uploaded an inter-author review draft with "Track Changes" enabled instead of a final paper. I assume this was a mistake as I am not a co-author. My role is review of your final product. Because this is not a final product, I will not review it further than the following:
Response: We are apologizing for that mistake and we have uploaded the final paper without "Track Changes" and indicated the changes with highlighted (Color) in major comments. We have improved the whole paper and corrected the English error from MDPI English editing company. If you think still have problems then we will improved according to your valuable suggestions.
Comments 2: The introductions provide sufficient background and include all relevant (Can be improved).
Response: Really thanks a lot for your technical suggestions; we have thoroughly revised the introduction with new additions at several places in order to address the aforementioned points (Page No. 2, line No.48-56, 61-71, 96-99 and 102-105).
Comments 3: Research design methods can be improved.
Response: Thank you very much for valuable comments; we revised the materials and methods section.
Comments 4: The results clearly presented (Must be improved).
Response: Thanks for fruitful advised; we have re written the result and discussion part with new additions, and improved English language (Page No. 3, line No.113-116, 134-137; Page No. 9, line No. 302-305 and Page No. 11, line No. 376-377).
Comments 5: The conclusions supported by the results (Must be improved).
Response: Thanks for your pertinent suggestion; we have re written the conclusion (Page No. 17, line No. 540-551).
Comments 6: The best I can understand from what I was sent is that the English is still problematic for an international audience.
Response: We have removed the English grammar errors and improved whole paper according to the (English Editing Company of MDPI by English native speaker).
Comments 7: The last sentence of the abstract appears to be a random comment rather than based on this work.
Response: Thanks for valuable comment; we have rewritten the last sentence of abstract (Page No.1, line No. 36-39. Finally we are grateful for your technical and fruitful suggestions, which have helpful for improved our paper.